# Different Chemical Forms of Thiamine, Riboflavin, and Folate in Human Milk as a Function of Lactation Stages—A Cohort Study on Breastfeeding Women from Beijing

**DOI:** 10.3390/nu17040624

**Published:** 2025-02-09

**Authors:** Ye Wang, Xinxin Xing, Xiangnan Ren, Shan Jiang, Zhenyu Yang, Jianqiang Lai

**Affiliations:** 1National Institute for Nutrition and Health, Chinese Center for Disease Control and Prevention, Beijing 100050, China; wangye@ninh.chinacdc.cn (Y.W.); tao280441@163.com (X.X.); renxn@ninh.chinacdc.cn (X.R.); jiangshan@ninh.chinacdc.cn (S.J.); yangzy@ninh.chinacdc.cn (Z.Y.); 2Key Laboratory of Human Breast Milk Science, Chinese Center for Disease Control and Prevention, Beijing 100050, China

**Keywords:** human milk, thiamine, riboflavin, folate, cohort

## Abstract

**Background:** The function and bioavailability of water-soluble vitamins in human milk (HM) is contingent upon their specific molecular configurations. This study aims to investigate the concentrations of different forms of thiamine, riboflavin, and folate in HM and to elucidate the temporal variations of these nutrients across different stages of lactation. **Methods**: A cohort of 35 healthy mother–infant pairs from Beijing was recruited, and 214 HM samples were collected. The concentrations of water-soluble vitamins in these samples were analyzed using high-performance liquid chromatography-tandem mass spectrometry (HPLC-MS/MS). A mixed linear regression model was employed to examine the relationship between HM vitamin levels and lactation stages. **Results**: This study analyzed the concentrations of free thiamine, thiamine monophosphate (TMP), thiamine pyrophosphate (TPP), free riboflavin, flavin adenine dinucleotide (FAD), flavin mononucleotide (FMN), 5-methyl-tetrahydrofolate (5-MTHF), tetrahydrofolate (THF), 5-formyl-tetrahydrofolate (5-fTHF), 5,10-methenyl-tetrahydrofolate (5,10-MTHF), and unmetabolized folic acid (UMFA) at various lactation stages (0–7 days, 15 days, 30 days, 60 days, 90 days, 120 days, 150 days, and 180 days). Free thiamine concentrations increased from colostrum to 180 days, while total thiamine rose during the first month and then stabilized. Free and total riboflavin levels remained relatively constant throughout lactation. Free and total folate concentrations peaked at 90 days and subsequently declined. Significant correlations were observed between follow-up time and changes in free thiamine, free folate, and total folate concentrations over 180 days. **Conclusions:** This study provides detailed data on the concentrations and trends of free and total thiamine, riboflavin, and folate in HM from 0 to 180 days postpartum, highlighting the dynamic nature of vitamin concentrations in HM. No deficiencies in these HM vitamins were detected in the surveyed population. Future further research will be conducted to reveal the correlation between different forms of water-soluble vitamins in HM and dietary factors.

## 1. Introduction

Human milk (HM) is the best food for infants [1]. HM contains a spectrum of water-soluble vitamins, including thiamine, riboflavin, and folate, which are critically involved in fundamental biological processes such as carbohydrate metabolism, nucleic acid synthesis, and amino acid homeostasis [2,3,4]. The concentrations of these vitamins exhibit dynamic variations across lactation stages to meet the physiological demands of infant development [5,6]. Consequently, precise quantification of vitamin profiles in HM throughout lactation is essential for establishing infant dietary reference intakes (DRIs) and optimizing breastfeeding practices.

Water-soluble vitamins in HM encompass a variety of forms. Thiamin in HM primarily consists of three forms: free thiamine, thiamine monophosphate (TMP), and thiamine pyrophosphate (TPP) [7]. For riboflavin, it mainly includes free riboflavin, flavin adenine dinucleotide (FAD), and flavin mononucleotide (FMN) [8,9]. HM folate naturally exists in its reduced form, mainly including 5-methyl-tetrahydrofolate (5-MTHF), tetrahydrofolate (THF), 5-formyl-tetrahydrofolate (5-fTHF), and 5,10-methenyl-tetrahydrofolate (5,10-MTHF), with trace amounts of oxidized unmetabolized folic acid (UMFA) originating from maternal supplementation or fortified food consumption [10,11]. Notably, each reduced folate species exhibits both free and protein-bound configurations through phosphate linkages.

The bioavailability of water-soluble vitamins is contingent upon their specific molecular configurations. Thiamine pyrophosphate (TPP) serves as an essential cofactor in decarboxylation reactions, FAD and FMN function as redox coenzymes in electron transport systems [7,8,9], while 5-MTHF as the predominant circulatory folate form, mediates one-carbon transfer reactions critical for nucleotide biosynthesis and epigenetic regulation [10,11]. The speciation patterns of these vitamins in HM are modulated by maternal determinants, including diet, nutritional status, genetic polymorphisms, and environmental exposures. Recent advances in analytical methodologies, particularly high-performance liquid chromatography-tandem mass spectrometry (HPLC-MS/MS), have enabled precise quantification of vitamin speciation profiles, thereby elucidating the complex interplay between maternal factors and HM vitamin composition

Despite their physiological significance, comprehensive characterization of vitamin speciation in HM remains limited. Our team conducted a longitudinal cohort study employing advanced HPLC-MS/MS to systematically quantify thiamine, riboflavin, and folate derivatives in HM samples collected across lactation stages, aiming to provide critical insights into the dynamic speciation patterns of water-soluble vitamins during lactation and grasp the nutritional status of HM in the surveyed subjects.

## 2. Materials and Methods

### 2.1. Study Design and Study Population

A prospective longitudinal cohort study was conducted at the Maternal and Child Health Hospital of Pinggu District, Beijing, China, between 2021 and 2023. Thirty-five pregnant women aged 20–40 years were enrolled during their third trimester (gestational weeks 32–36). Inclusion criteria required the following: (1) uncomplicated singleton pregnancy, (2) vaginal delivery at full-term (≥37 weeks), (3) absence of gestational complications (hypertension, diabetes, anemia), (4) commitment to exclusive breastfeeding for 6 months postpartum. Exclusion criteria included the following: (1) maternal smoking, alcohol consumption, or exposure to environmental toxicants, (2) hormonal therapy during pregnancy, (3) preterm delivery (<37 weeks) or failure to maintain exclusive breastfeeding. HM samples were collected longitudinally at eight time points: 0–7 days (colostrum), and 15 ± 1, 30 ± 3, 60 ± 3, 90 ± 3, 120 ± 3, 150 ± 3, and 180 ± 3 days postpartum. Participants completing ≥ 6 follow-ups were retained for analysis, yielding 28 maternal–infant dyads with complete datasets (see Figure 1).

This study was registered at the Chinese Clinical Trial Center (ChiCTR2000041342), approval date 23 December 2020. It has been approved by the Ethics Committee of the National Institute for Nutrition and Health, China CDC, approval code 2020-23, approval date 7 September 2020. Informed consent was signed by all included individuals.

### 2.2. Sample Collection

HM samples were collected at the hospital (colostrum) or the lactation mother’s home (transition milk and mature milk) between 9:00 am and 11:00 am on the day of the follow-up. Lactating mothers used electric breast pumps to empty one side of their breasts. The amount of colostrum collected was 5–15 mL, and the amount of transitional and mature milk collected was 20 mL. The remaining HM was returned to the infant or given to the researchers according to the personal wishes of the lactation mother.

At the hospital, the nurse assisted the mother in collecting HM and immediately homogenized (shaking 10 times up and down) and separated the milk samples, which were then stored in a −80 °C refrigerator. At the lactating mother’s home, HM was collected by the mother themself and immediately placed in a 4 °C refrigerator. The researchers collected the HM samples from the home within 2 h and transported them to the hospital under 4 °C within 2 h. Upon arrival at the hospital, the milk samples were immediately homogenized, separated, and stored in a −80 °C refrigerator.

### 2.3. Information Collection

Demographic information (civil status, educational background) was collected via self-reported questionnaires during enrollment, with nurses’ help in the hospital. Health data (maternal age at birth, pre-pregnancy BMI, gestational weight gain, mode of birth, intake of dietary supplements during pregnancy and lactation, and birth weight) were extracted from medical records.

### 2.4. Materials

Thiamine, TMP, TPP, riboflavin, FAD, FMN, 5-MTHF, THF, 5-fTHF, 5,10-MTHF, and UMFA were analyzed by using high-performance liquid chromatography-tandem mass spectrometry (HPLC-MS/MS). The standards used during the analyses were as follows: vitamin B1-^13^C_4_ (>98% purity, Medical Isotopes, Inc., Pelham, NH, USA), vitamin B2-^13^C_4_,^15^N_2_ (Cerilliant, Round Rock, TX, USA), thiamine monophosphate (Sigma–Aldrich, St. Louis, MO, USA), thiamine pyrophosphate (Sigma–Aldrich), flavin adenine dinucleotide (Sigma–Aldrich, USA), flavin mononucleotide (Sigma–Aldrich), folic acid (Sigma–Aldrich), THF (Sigma–Aldrich), 5-fTHF (Sigma–Aldrich), folic acid-(glutamic acid-^13^C_5_) (internal standard, Sigma–Aldrich), and 5-MTHF (TRC), 5,10-MTHF (TRC), 5-methyltetrahydrofolic acid-(glutamic acid-^13^C_5_) (internal standard, Cambridge Isotope Laboratories, Inc., Tewksbury, MA, USA). Methanol, acetonitrile, ethanol, hexane, ethyl acetate, formic acid, and ammonium acetate of HPLC-MS/MS grade were purchased from Thermo Fisher Scientific (Fair Lawn, NJ, USA). All other chemicals were of analytical grade and purchased locally. Ultra-pure water was Watson’s distilled water.

### 2.5. Analysis Procedure

For thiamine, TMP, TPP, riboflavin, FAD, FMN, UMFA, and free 5-MTHF, THF, 5-fTHF, 5,10-MTHF, sample preparation consisted of a simple protein precipitation procedure. A total of 90 μL of the HM samples were first mixed with 10 μL of internal standard solution (Vitamin B1-^13^C_4_, Vitamin B2-^13^C_4_,^15^N_2_, TMP, TPP, FAD, FMN, Folic acid-(glutamic acid-^13^C_5_), 5-Methyltetrahydrofolic acid-(glutamic acid-^13^C_5_), and Methyl-D3-malonic acid) which were prepared in methanol. Then, 300 μL of methanol was added to the mixture before vortexing for 3 min. Next, samples were centrifuged for 10 min at 15,000 rpm at room temperature. A total of 150 μL of the supernatant was transferred into a 96-well plate and covered with a silicone mat. A total of 3 μL was injected into the HPLC-MS/MS system.

For total 5-MTHF, THF, 5-fTHF, and 5,10-MTHF, the samples were thawed in a dark room using a warm water bath. A total of 100 µL of the sample was transferred into a 2 mL centrifuge tube, and 10 µL of the internal standard solution was added (200 ng/mL, prepared in 20% methanol–water). Then, 500 µL of diluent was added (0.1 mol/L sodium acetate solution containing 1% ascorbic acid and 0.5% thioglycolic acid, adjusted to pH 4.5 with formic acid), and then 20 µL of rat serum was added (treated with activated carbon). Incubate at 37 °C for 2 h. After incubation, 200 µL of 0.2% ammonia water and 0.8 mL of dichloromethane was added, vortexed, and shaken for 2 min to degrease and centrifuged at 15,000 rpm for 10 min. The supernatant to be purified was loaded on the solid phase extraction plate, eluted with 100 µL of pure water under negative pressure, and then eluted twice with 100 µL of 2% formic acid methanol (containing 1% diluent). The eluate was dried under nitrogen at 50 °C, reconstituted with 100 µL of pure water, shaken and mixed, and centrifuged at 15,000 rpm for 10 min. The supernatant was injected into the LC-MS/MS system with a sample volume of 10 µL. All samples were processed under red light.

Recovery rates of thiamine, TMP, TPP, riboflavin, FAD, FMN, 5-MTHF, THF, 5-fTHF, 5,10-MTHF, and UMFA ranged from 95% to 110%.

### 2.6. Statistical Analyses

Total thiamine represents the sum of free thiamine, TMP, and TPP [9]. Total riboflavin represents the sum of riboflavin, FAD, and FMN [9,12]. Free folate represents the sum of UMFA, free 5-MTHF, THF, 5-fTHF, and 5,10-MTHF. Total folate represents the sum of UMFA, total 5-MTHF, THF, 5-fTHF, and 5,10-MTHF [10].

Statistical description and analysis were performed using SAS 9.4. The basic characteristic data of the research object is described by the adoption rate and composition ratio. Normality testing was performed on the data. For data that conforms to a normal distribution, it is represented by the mean ± sd, and for non-normal data, it is represented by M (P_25_–P_75_). Our study also uses a mixed linear regression model to investigate the relationship between HM vitamin status and lactation stages (follow-up time) and covariates (neonatal gender, birth weight, gestational weight gain, etc.). The random effects are the study objects, while the fixed effects are the follow-up time. Parameters are estimated using the SAS software MIXED procedure.

## 3. Results

### 3.1. Characteristics of the Study Population and Human Milk Collection

The basic information of the pregnant women included in the study is shown in Table 1. Ultimately, 28 participants completed the follow-up, of which 21 participants completed all 8 follow-ups, 4 participants completed 7 follow-ups, and the other 3 participants completed 6 follow-ups. A total of 214 HM samples were collected. All mothers were of Han Nationality and adhered to exclusive breastfeeding until the end of the follow-up period. All pregnancies were at term.

### 3.2. Vitamin Concentration and Change Trend in HM Milk

The vitamin values measured at different follow-up time points are shown in Figure 2 and Appendix A. A mixed-effects linear model single-factor analysis was conducted to analyze the relationship between follow-up time and the concentration of various vitamins (Table 2).

The concentration of free thiamine showed an upward trend, from 1.11 μg/L (colostrum) increased to 25.59 μg/L (180 d), and had a significant correlation between follow-up time and content change within 180 days (*P_all_* < 0.0001). TMP is the major thiamine form in HM. Affected by this, the concentration of total thiamine showed an upward trend during the first month (from 33.96 μg/L to 185.47 μg/L), after which the concentration remained relatively stable. TPP concentrations are steadily maintained at a relatively low level.

The concentration of free and total riboflavin did not vary significantly during the lactation period (*P_all_* > 0.05). The concentration of free riboflavin from 120 to 150 days varied significantly over time, possibly due to the sample size. Since FAD predominates in HM riboflavin, the content of total riboflavin is much higher than free riboflavin (about 20 times higher). The content of FAD is also relatively low, increasing from 4.72 μg/L (colostrum) to 13.79 μg/L (60 days) and then remaining relatively stable.

5-MTHF is the major form of HM folate. Since the content of all types of free reduced folate other than free 5-MTHF is lower than the detection limit, the content of free folate is equivalent to the sum of free 5-MTHF and UMFA. The content of free folate reached its highest level at 90 days (16.61 μg/L) and then gradually decreased; there was a significant correlation between follow-up time and changes in free folate concentrations (*P_all_* < 0.0001). For total folate, the content reached its highest level at 90 days (49.88 μg/L) and then gradually decreased to 40.31 μg/L at 180 days. The correlation between follow-up time and changes in total folate concentrations was also significant (*P_all_* < 0.0001).

Based on factors such as mother education level, household income, gestational weight gain, offspring gender, and offspring birth weight, different groups of lactating mothers with varying characteristics were categorized, but no significant differences were observed between the groups (See Appendix A).

Correlation Between Different Maternal and Infant Characteristics and the Trend of Vitamin Concentration Changes in HM

A mixed-effects linear model analysis was conducted on the basic characteristics of mothers and infants and the trend of changes in water-soluble vitamins. The results showed that there were no significant differences in the change trends of free or total vitamin concentration among different birth weights (SGA vs. AGA vs. LGA), infant gender (Male vs. Female), GWG (Inadequate vs. Adequate vs. Excessive), and household income (<20,000 CNY vs. ≥20,000 CNY) sub-groups (*p* ≥ 0.05).

## 4. Discussion

### 4.1. Thiamine

TMP accounts for over 70% of the total thiamine content, while free thiamine constitutes ≤ 25% (10.5–13.3 μg/L) [9]. These molecular species undergo interconversion via phosphorylation-dephosphorylation reactions. Our previous study [8] of 1778 HM samples across lactation stages demonstrated progressive increases in free thiamine concentrations: colostrum (5.0 μg/L), transitional milk (6.7 μg/L), early mature milk (15–180 days; 21.1 μg/L), and late mature milk (>180 days; 40.7 μg/L). Compared to this study, Ren’s [8] study showed significantly higher thiamine concentration in colostrum, possibly due to most colostrum samples in our study being collected from the first 3 days. The concentration of early mature milk was closer to the average value of all mature milk in our study (21.3 μg/L). A cross-sectional study conducted by Shi et al. [13] in Inner Mongolia, China, indicated that the free thiamine concentrations in colostrum, transitional milk, and mature milk were 65 μg/L, 52 μg/L, and 63 μg/L, respectively. From a methodological perspective, this study likely measured total thiamine, with higher levels observed in colostrum compared to our study. Another study conducted by Xue et al. [12] in China showed that the free thiamine concentrations in HM at 5–11 d, 12–30 d, 31–60 d, and 61–120 d were 31.3 μg/L, 50.7 μg/L, 42.8 μg/L, and 56.5 μg/L, respectively. These levels of free thiamine are significantly higher than those in our study, possibly due to different pre-treatment before testing, in which some amylase (taka-amylase) may convert thiamine phosphate esters into the parent molecule [14]. In other words, Xue’s study actually reported total thiamine rather than free thiamine. Therefore, the total thiamine levels in our study in both early and mature milk are higher than those reported by Xue. Another study showed that only the free thiamine concentration increased during lactation, suggesting that the phosphorylation or hydrolysis mechanism of thiamine may be more active during early lactation [9].

While TPP’s role as a coenzyme in carbohydrate metabolism (transketolase, α-ketoglutarate dehydrogenase, pyruvate dehydrogenase) is well-established [7,15,16], the physiological functions of TMP and free thiamine in HM remain poorly characterized. In tissues such as kidney or embryonic tissues, the content of TMP has been found to be significantly higher than that of other thiamine derivatives [17]. Studies have shown that the transport of TPP and TMP from maternal blood to HM relies on the reduced folate carrier (RFC), while the transport of free thiamine occurs through thiamine transporters 1 and 2 (THTR-1, THTR-2) [18,19]. Stuetz et al. [7] indicate that a high ratio of TMP to total thiamine is associated with low whole blood TPP and insufficient total thiamine concentration in HM. In our study, the proportion of TMP in HM generally exceeded 90%, which is much higher than that reported in Stuetz’s study and may warrant further investigation into maternal thiamine nutritional status and its impact on HM composition.

### 4.2. Riboflavin

The forms of riboflavin in human milk mainly include free riboflavin, FAD, and FMN, of which the content of FAD accounts for 85–94% of the total riboflavin content, and free riboflavin accounts for 6–15% [8,9,12]. Ren et al. [8] reported that the free riboflavin concentrations in colostrum, transitional milk, early mature milk, and late mature milk were 29.3 μg/L, 40.6 μg/L, 33.6 μg/L, and 29.6 μg/L. The highest concentration of transitional milk is the same as that in our study. Shi et al. [13] reported that the concentrations of colostrum, transitional milk, and mature milk were 169 μg/L, 176 μg/L, and 137 μg/L, significantly lower than our study. Xue et al. [12] also reported the free riboflavin concentrations in HM at 5–11 d, 12–30 d, 31–60 d, and 61–120 d were 208 μg/L, 202 μg/L, 119 μg/L, and 136 μg/L, significantly higher than our and Ren’s report. The reason may also be due to using taka-amylase during pre-treatment before testing, which means that Xue’s study may actually report the total riboflavin content.

The riboflavin family plays a role in mitochondrial electron transport and energy metabolism, cellular stress response modulation, and immune system development. HM is crucial for the development of the infant immune response, with FAD and FMN playing significant roles. Riboflavin is transported by a group of transporters in the solute carrier family SLC52 and riboflavin transporter 3 (RFVT3) [4,20]. Current evidence suggests that riboflavin speciation in HM is primarily influenced by maternal supplementation (usually in free form) practices rather than systemic riboflavin status or health parameters [9]. Since the participants in this study did not take any dietary supplements after the second trimester, the ratio of FAD to free riboflavin was relatively consistent.

### 4.3. Folate

Folate in HM is naturally mainly in reduced form, including 5-MTHF, THF, 5-fTHF, and 5,10-MTHF [21]. 5-MTHF is the predominant form of folate in HM. Folate in dietary supplements or fortified foods is mainly present in oxidized form (i.e., folic acid, FA). FA cannot be directly utilized by humans and must be converted into THF under dihydrofolate reductase (DHFR) in the liver, then converted into 5,10-MTHF under the action of methylenetetrahydrofolate dehydrogenase (MTHFD), and then converted into 5-MTHF under the action of methylenetetrahydrofolate reductase (MTHFR) [22]. However, the reduction ability of DHFR in humans is easily saturated. If FA exceeds the reduction ability of DHFR, it may lead to an insufficient reduction of FA. These accumulated FA in the human body, and HM is called unmetabolized folic acid (UMFA) [23].

There are many reports on the total folate concentrations in HM. Xue [12] reported that the total folate concentrations in HM during 12–30 d, 31–60 d, 61–120 d, and 121–240 d were 7.30, 23.90 ng/mL, 24.40 ng/mL, 24.2 ng/mL, and 23.30 ng/mL, respectively. Nguyen [24] reported that the total folate concentration in mature milk in Chinese mothers was 32.3 ± 13.1 ng/mL. Su [16] was the first to report the content of different forms of folate in Chinese HM, which used LC-MS/MS method, reported the concentrations of reduced folate in HM at 40–45 d, 200–240 d, and 300–400 d, were 54.65 ng/mL, 51.29 ng/mL, and 52.71 ng/mL respectively, and 54.65 ng/mL, 56.77 ng/mL, 55.14 ng/mL for total folate. The concentration and temporal variation of folate in HM in our study were similar to those of Su’s report but higher than many previous reports on the folate content of Chinese HM [12,13,24]. It may be due to the fact that we and Su had converted all bound folate to free form during preprocessing (use rat serum), resulting in higher concentration.

Page [25] found that due to the high intake of folic acid (from dietary supplements and fortified foods), the UMFA level in HM has reached as high as 21 ng/L in Canada. In contrast, UMFA concentration in our study has been quietly low, possibly because none of the mothers in our cohort took folic acid supplements after the 2nd trimester (after 30 weeks) or after delivery. A total of 61% of mothers in our cohort took folic acid tablets (Elevit, Bayer, Germany, 0.4 mg/tablet, 1 tablet per day) from pre-pregnancy to 2nd trimester, while the rest did not take folic acid supplements. However, there was no significant difference in free 5-MTHF concentration between these two groups of mothers during all lactation stages after delivery.

### 4.4. Reasons for Using Cohort Study to Investigate the HM

Existing data on the HM vitamin content mostly come from cross-sectional studies, which do not distinguish between lactation stages, assigning colostrum, transitional milk, and mature milk into the same category [26], or classifying lactation stages too broadly [8], masking the dynamic changes in related vitamin content. Our study adopted a cohort study design to investigate the composition of HM at different lactation stages, which can better address this issue by collecting multiple milk samples from the same breastfeeding mother at designated follow-up times and conducting laboratory analysis to provide more accurate data on HM composition [27,28,29]. However, even with the use of cohort studies, some HM study cohorts still have issues with overly dispersed follow-up times or excessively long intervals [30]. Therefore, our study specifically designed a unified follow-up time to accurately verify the role of time factors in nutrient changes.

### 4.5. Limitations

Firstly, due to the use of a cohort study design in this research, the sample size is relatively limited, and the relevant nutrient contents may not be representative of the entire China. Secondly, due to the COVID-19 pandemic, follow-up and data collection have been challenging in hospitals, resulting in this study lacking dietary intake information (only dietary supplement intake was recorded), which may potentially mask the interaction between diet and HM composition. Our future research will focus on investigating dietary patterns during pregnancy and lactation. Thirdly, we lack an understanding of the clinical effects of different chemical forms of water-soluble vitamins studied in this research, especially thiamine and riboflavin. More observational studies and animal experiments need to be conducted in the future to observe the correlation between the TMP/TTP ratio and infant development indicators.

## 5. Conclusions

Our study is the first to adopt a rigorous cohort study method, reporting the levels and trends of free and total thiamine, riboflavin, and folate at different lactation stages in China, covering 8 follow-up times. We utilized more advanced laboratory analysis methods, which enabled us to better understand the regularities of the composition of HM in China. Overall, there is no deficiency related to HM vitamins in the surveyed areas. However, a comparison with studies from other countries suggests that more attention may need to be paid to the dietary intake of water-soluble vitamins among these lactating mothers. Due to the fact that diet can significantly influence the water-soluble vitamins in HM, the lack of dietary data has constrained further research on these components in our study. Furthermore, findings on different forms of vitamins in HM can be applied in HM banks in the future; providing more appropriate HM for infants at different lactating stages and identifying unconventional substances (such as excessive UMFA) in HM may help improve the feeding quality of infants in special situations.

## Figures and Tables

**Figure 1 nutrients-17-00624-f001:**
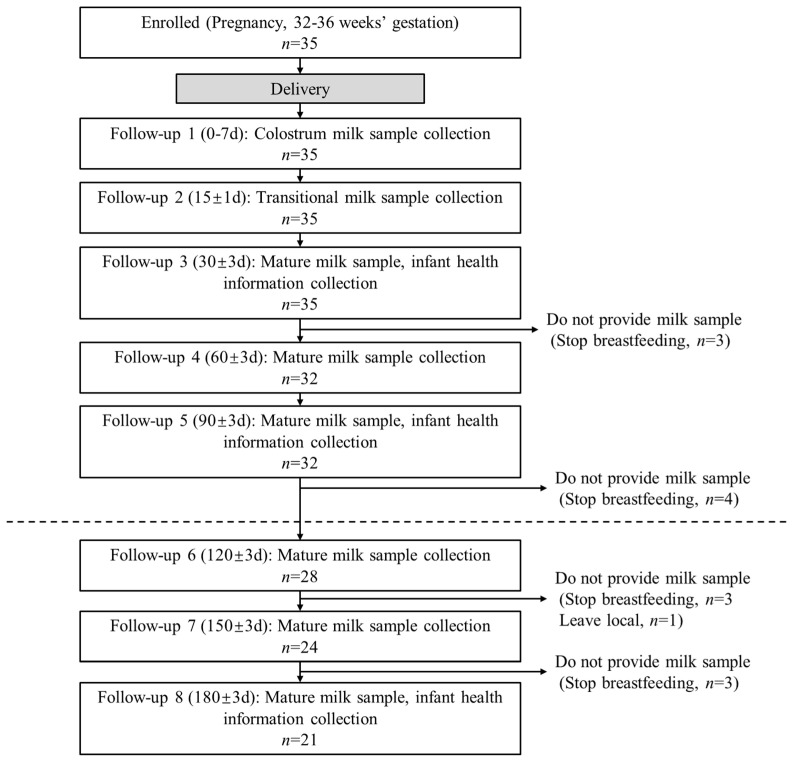
Participant flow diagram. Participants completing ≥ 6 follow-ups were retained for analysis (dash line).

**Figure 2 nutrients-17-00624-f002:**
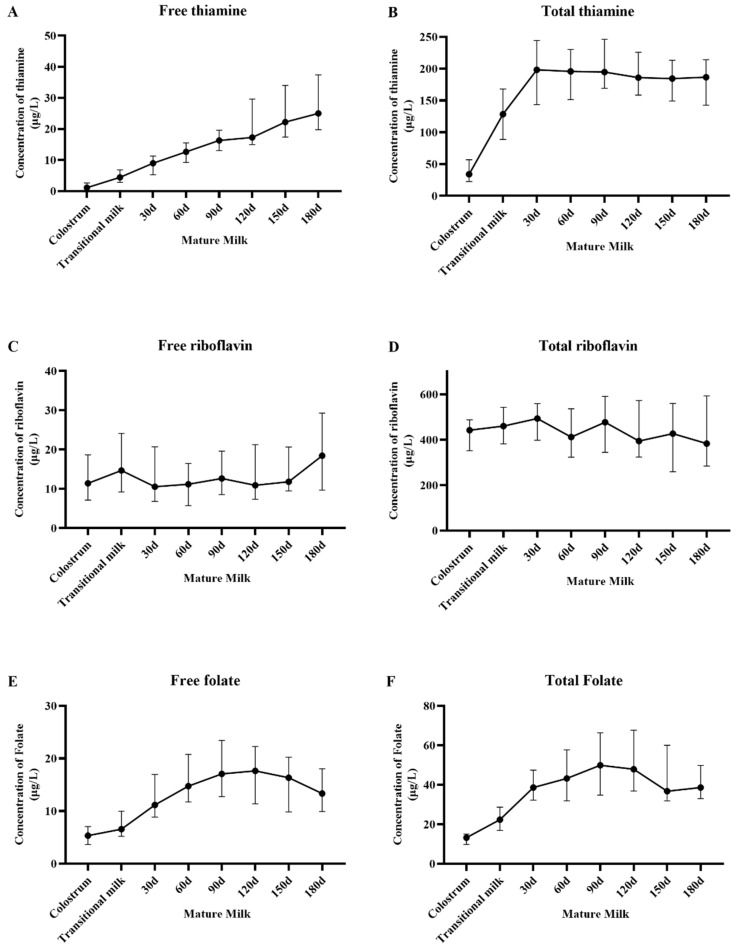
Changes in HM vitamin concentration during the follow-up period (IQR). (**A**) Free thiamine, (**B**) Total thiamine, (**C**) Free riboflavin, (**D**) Total riboflavin, (**E**) Free folate, and (**F**) Total folate.

**Table 1 nutrients-17-00624-t001:** Characteristics of participant mothers and corresponding infants.

	Participant (*n* = 28)
Age, years	28.7 ± 3.2
Education, *n* (%)	
Senior middle school and below	5 (18%)
College and above	23 (82%)
Household income per year, CNY, *n* (%)	
<20,000	21 (75%)
≥20,000	7 (25%)
Pre-pregnancy BMI (kg/m^2^)	21.70 (20.14, 21.92)
Gestational weight gain (GWG) (kg)	15.29 ± 6.26
Inadequate GWG, *n* (%) *	2 (7%)
Appropriate GWG, *n* (%) *	8 (29%)
Excessive GWG, *n* (%) *	18 (64%)
Weeks of gestation at delivery, weeks **	40.4 ± 1.2
Offspring gender	
Male, *n* (%)	13 (46%)
Female, *n* (%)	15 (54%)
Offspring birth weight, g	3449.0 ± 436.3
SGA, *n* (%) ***	3 (11%)
AGA, *n* (%) ***	19 (68%)
LGA, *n* (%) ***	6 (21%)
Offspring birth length, cm	51.2 ± 1.5

* According to Chinese GWG standard; ** All pregnancies were at term; *** According to Chinese standard.

**Table 2 nutrients-17-00624-t002:** Trend of changes of HM water-soluble vitamin at different follow-up times. * The difference is significant.

	Free Thiamine	Total Thiamine	Free Riboflavin	Total Riboflavin	Free Folate	Total Folate
***P_(0–7d_* vs. *_14d)_***	0.1437	<0.0001 *	0.7798	0.2339	0.0796	0.0006 *
***P_(14d_* vs. *_30d)_***	0.1024	<0.0001 *	0.7851	0.5855	0.0006 *	<0.0001 *
***P_(30d_* vs. *_60d)_***	0.0348 *	0.9734	0.8708	0.0732	0.0151 *	0.2946
***P_(60d_* vs. *_90d)_***	0.0034 *	0.4343	0.8990	0.0829	0.0757	0.0330 *
***P_(90d_* vs. *_120d)_***	0.1463	0.3389	0.8713	0.2200	0.9104	0.8007
***P_(120d_* vs. *_150d)_***	0.0003 *	0.9694	0.0076 *	0.9249	0.0197 *	0.0174 *
***P_(150d_* vs. *_180d)_***	0.1616	0.8157	0.3059	0.6142	0.2698	0.7193
** *P_all_* **	<0.0001	0.5355	0.1710	0.8896	<0.0001	<0.0001

## Data Availability

The original contributions presented in the study are included in the article/Appendix A, further inquiries can be directed to the corresponding author.

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
