# Peer review of "Different Chemical Forms of Thiamine, Riboflavin, and Folate in Human Milk as a Function of Lactation Stages—A Cohort Study on Breastfeeding Women from Beijing"

_nutrients, 2025, doi:10.3390/nu17040624_

Round 1

Reviewer 1 Report

Comments and Suggestions for Authors

To the Authors

Review for the Ms ID Nutrients-3447820 entitled “Different form of thiamine, riboflavin and folate in Human Milk among different lactation stages -A Cohort Study in Beijing”. The rationale behind the present study is apparently based on the limited knowledge regarding the different forms of thiamine, riboflavin, and folate in human milk (HM). The laboratory work is to be appreciated for accuracy , several flaws can be recognizable.  Although the cohort design could be helpful in giving potential reference range for the free and total forms of the examined HM composition in Chinese breastfeeding mothers, several limitations hamper a generalization and a proper interpretation of the results

Major points

1.    Besides the results reported in Table 2, all the correlations between infant-mother pair characteristics and aumont of different chemical forms  of thiamine, riboflavin, and folate in HM as a function of lactational stages should be accurately reported (either as supplementary files or in the main file)

2.    The Discussion section should be fully re-edited, expanded and more focused on the interpretation of the results in terms of the biological relevance of the different chemical forms of the analyzed vitamins rather than a simple comparison with similar studies.

Minor points

1.    Title. I would suggest to rephrase the current title from the current one (Different form of thiamine, riboflavin and folate in Human Milk among different lactation stages -A Cohort Study in Beijing) to one like Different chemical forms of thiamine, riboflavin and folate in Human Milk as a function of lactation stages- A Cohort Study on breastfeeding women from Beijing”. Of course, this is just a personal and non-binding suggestion in order to improve the title clarity. However, it is clinical relevant to include that these are breastfeeding women who, if I well understand the statistical data in Table 1, have delivered at term gestation.

1.    Abstract, LL 17-23. Listing the results as a simple list of numbers is rather meaningless and confounding. Please rephrase the study findings with more attention to the readership comprehension.  

2.    Introduction. The introductory paragraphs should be shortened up in order to reach the key point of the study introduction (LL 37 onwards). Please pay particular attention to what is known about the biological and clinical relevance of the different chemical forms of the examined vitamins in HM.  

3.    Introduction. The paragraphs at LL 56-62 should rephrased as proper introductory statements while some of the comments could be either replaced in the Discussion section or removed

4.    Materials & Methods. The paragraphs at LL 68-71 should be conjugated in the plural form (“Mother was pregnant normally… The infant was full-term, delivered vaginally and is scheduled to be exclusively  breastfed in 6 months”).

5.    Materials & Methods. Verbs in the paragraphs in LL 75-77 should be conjugated in the past tense form (“All breastfeeding mothers are required … otherwise they will be considered for a withdrawal. Eventually the pair completed the study. (see Figure 1.))

6.    Materials & Methods, LL 91. (“At hospital, the nurse assists the mother in collecting HM, and immediately homogenizes). Please correct the sentence verb in the past tense, and change “homogenizes” in “homogenized”

7.    Materials & Methods, LL 100. “self-report” should be “self-reported”

8.    Discussion, LL 267 (“UMFA concentration in our study has been quietly low…”). The term “quietly” is a conceptual nonsense. Please clarify.

9.    Study Limitations, LL 286-292. In my personal judgment, the lack of knowledge regarding maternal dietary intakes is indeed a major study limitation. Please further expand on this point and clarify/justify.

10.  Study Limitations, LL 286-292- I would like to add to the limitations already self-declared by the Authors, the one regarding the lack of knowledge on the clinical effects of the different chemical forms of the water-soluble vitamins examined in the study. Please clarify/justify.

11.  References list. Just about one quarter (i.e. 6/23) of the cited references were published in the last 5 years. Please check and/or update the existing bibliography.   

12.  Table 1: “Gestational weight (kg)” I would assume to be meant as  “gestational weight gain”. Please correct or justify/clarify.

13.  Table 1: The gestational age reported in Table 1 would suggest that all the pregnancies were at term. Please clarify.

14.  Although the quality of the English does not prevent the interpretation of the study findings, I would suggest an extensive revision of language grammar and style for a better readability and communication clarity.

Comments on the Quality of English Language

Although the quality of the English does not fully prevent the interpretation of the study findings, I would highly recommend an extensive revision of language grammar and style for more clearly express the current research. 

Author Response

Comments 1: Besides the results reported in Table 2, all the correlations between infant-mother pair characteristics and aumont of different chemical forms  of thiamine, riboflavin, and folate in HM as a function of lactational stages should be accurately reported (either as supplementary files or in the main file).

Response 1: Thank you for pointing this out. I agree with this comment. Therefore, I have added detailed comparisons of infant-mother pair characteristics and the amounts of different chemical forms of thiamine, riboflavin, and folate in HM. Please refer to the revised manuscript L195-198 and supplementary table 2a-2e for details.

Comments 2: The Discussion section should be fully re-edited, expanded and more focused on the interpretation of the results in terms of the biological relevance of the different chemical forms of the analyzed vitamins rather than a simple comparison with similar studies.

Response 2: Agree. As required, I rewrote a significant portion of the discussion section. Change can be found in the revised manuscript L243-254, L268-276

Comments 3: Title. I would suggest to rephrase the current title from the current one (Different form of thiamine, riboflavin and folate in Human Milk among different lactation stages -A Cohort Study in Beijing) to one like “Different chemical forms of thiamine, riboflavin and folate in Human Milk as a function of lactation stages- A Cohort Study on breastfeeding women from Beijing”. Of course, this is just a personal and non-binding suggestion in order to improve the title clarity. However, it is clinical relevant to include that these are breastfeeding women who, if I well understand the statistical data in Table 1, have delivered at term gestation.

Response 3: Agree. I have changed the title to “Different chemical forms of thiamine, riboflavin and folate in Human Milk as a function of lactation stages- A Cohort Study on breastfeeding women from Beijing”

Comments 4: Abstract, LL 17-23. Listing the results as a simple list of numbers is rather meaningless and confounding. Please rephrase the study findings with more attention to the readership comprehension.

Response 4: Agree. I have re-write this part. See L15-L25

Comments 5: Introduction. The introductory paragraphs should be shortened up in order to reach the key point of the study introduction (LL 37 onwards). Please pay particular attention to what is known about the biological and clinical relevance of the different chemical forms of the examined vitamins in HM.

Response 5: Agree. I reduced the less relevant parts as required and added research background and progress of these three vitamins as requested by other reviewers See L32-39

Comments 6: Introduction. The paragraphs at LL 56-62 should rephrased as proper introductory statements while some of the comments could be either replaced in the Discussion section or removed

Response 6: Agree. I have re-write this part, and replaced to the Discussion section. See L310-320

Comments 7: Materials & Methods. The paragraphs at LL 68-71 should be conjugated in the plural form (“Mother was pregnant normally… The infant was full-term, delivered vaginally and is scheduled to be exclusively  breastfed in 6 months”)

Response 7: Agree. I have re-write this part. See L68-79

Comments 8: Materials & Methods. Verbs in the paragraphs in LL 75-77 should be conjugated in the past tense form (“All breastfeeding mothers are required … otherwise they will be considered for a withdrawal. Eventually the pair completed the study. (see Figure 1.))

Response 8: Agree. I have changed these words. See L68-79

Comments 9: Materials & Methods, LL 91. (“At hospital, the nurse assists the mother in collecting HM, and immediately homogenizes). Please correct the sentence verb in the past tense, and change “homogenizes” in “homogenized”

Response 9: Agree. I have re-write this word. See L98

Comments 10: Materials & Methods, LL 100. “self-report” should be “self-reported”

Response 10: Agree. I have re-write this word. See L102

Comments 11: Discussion, LL 267 (“UMFA concentration in our study has been quietly low…”). The term “quietly” is a conceptual nonsense. Please clarify.

Response 11: Due to the widespread use of folic acid dietary supplements, a significant amount of UMFA has been found in HM in recent years, which is not supposed to be present in large quantities. Therefore, this statement is used to emphasize the low level of UMFA in our study. The Canadian study has been added as background information. See L300-302

Comments 12: Study Limitations, LL 286-292. In my personal judgment, the lack of knowledge regarding maternal dietary intakes is indeed a major study limitation. Please further expand on this point and clarify/justify.

Response 12: Agree. I have re-write this part. See L324-352

Comments 13: Study Limitations, LL 286-292- I would like to add to the limitations already self-declared by the Authors, the one regarding the lack of knowledge on the clinical effects of the different chemical forms of the water-soluble vitamins examined in the study. Please clarify/justify.

Response 13: Agree. I have re-write this part. See L324-352

Comments 14: References list. Just about one quarter (i.e. 6/23) of the cited references were published in the last 5 years. Please check and/or update the existing bibliography.

Response 14: Due to the scarcity of research on different forms of water-soluble vitamins in HM, there are indeed relatively few references within the past 5 years. I have reviewed relevant literature again and tried to include some research from relative fields.

Comments 15: Table 1: “Gestational weight (kg)” I would assume to be meant as  “gestational weight gain”. Please correct or justify/clarify.

Response 15: Agree. I have changed these words. See Table1

Comments 16: Table 1: The gestational age reported in Table 1 would suggest that all the pregnancies were at term. Please clarify.

Response 16: Agree. I have changed these words. See Table1

Comments 17: Although the quality of the English does not prevent the interpretation of the study findings, I would suggest an extensive revision of language grammar and style for a better readability and communication clarity.

Response 17: Agree. The grammar of the entire text has been revised.

Reviewer 2 Report

Comments and Suggestions for Authors

Before this manuscript can be considered for publication in Nutrients, I encourage the authors to make some revisions. These are my suggestions:

You should start your abstract by providing a background statement to justify the need for carrying out your study and then the study’s objectives need to be clarified. Finally, some directions for further investigation should be provided.

References must be formatted according to the journal’s guidelines.

The Introduction is very brief and poor. Please, provide more information on the topics addressed in your study and you have to make clear the need to conduct the study. The inclusion of tables and illustrations in this section is recommended.

Line 65: Why 35 pregnant women were recruited from only one center? How do you consider this sample representative and adequate?

Explain better how the information was collected (section 2.3).

The Results are adequate but in the Discussion, more is expected. More studies from other studies conducted in other regions should be analyzed and the limitations subsection should be clarified and better discussed.

The Conclusions also need to be improved and aligned with the revised abstract. Future perspectives are missing.

Author Response

Comments 1: You should start your abstract by providing a background statement to justify the need for carrying out your study and then the study’s objectives need to be clarified. Finally, some directions for further investigation should be provided.

Response 1: Thank you for pointing this out. Agree. I have re-write this part. See L9-28 and L324-346

Comments 2: References must be formatted according to the journal’s guidelines.

Response 2: Agree. I have formatted references. See L361-429

Comments 3: The Introduction is very brief and poor. Please, provide more information on the topics addressed in your study and you have to make clear the need to conduct the study. The inclusion of tables and illustrations in this section is recommended.

Response 3: Agree. I have re-write this part. See L50-65

Comments 4: Line 65: Why 35 pregnant women were recruited from only one center? How do you consider this sample representative and adequate?

Response 4: Originally, we planned to establish research cohorts in multiple regions of China. However, due to the COVID-19 pandemic, we only established a cohort in Beijing from 2021 to 2022. Given the issues of representativeness and small sample size, since 2023, we have established new research cohorts in multiple regions of China, and will continue to report on them in future studies.

Comments 5: Explain better how the information was collected (section 2.3).

Response 5: I have added some descriptions. Our information collection is indeed solely based on the questionnaire filled out during enrollment and the hospital's HIS system. If necessary, please inform me of any additional information that needs to be provided See L102

Comments 6: The Results are adequate but in the Discussion, more is expected. More studies from other studies conducted in other regions should be analyzed and the limitations subsection should be clarified and better discussed.

Response 6: Agree. I have re-write this part. See L243-254 and L268-276

However, due to the scarcity of research on different forms of water-soluble vitamins in HM, there are indeed relatively few reports from global.  

Comments 7: The Conclusions also need to be improved and aligned with the revised abstract. Future perspectives are missing.

Response 7: Agree. I have re-write this part. See L324-332 and L336-346

Reviewer 3 Report

Comments and Suggestions for Authors

The presented manuscript of Dr. Jianqiang team has evaluated the thiamine, riboflavin and folate dynamics in human breastmilk over the first six months post delivery that is a subject of intense scientific interest. The methodology is sound and the results are solid. The paper provide correlation analysis with certain anthropometric parameters in both the mother and the infant. The discussion should be expanded especially in regards to comparison with untargeted metabolomics studies of breastmilk. Despite its limitations (relatively small number of participant in the cohort) the presented paper would be of high interest for the Nutrients auditory.

Author Response

Comments 1: The presented manuscript of Dr. Jianqiang team has evaluated the thiamine, riboflavin and folate dynamics in human breastmilk over the first six months post delivery that is a subject of intense scientific interest. The methodology is sound and the results are solid. The paper provide correlation analysis with certain anthropometric parameters in both the mother and the infant. The discussion should be expanded especially in regards to comparison with untargeted metabolomics studies of breastmilk. Despite its limitations (relatively small number of participant in the cohort) the presented paper would be of high interest for the Nutrients auditory.

Response 1: Thank you for pointing this out. Unfortunately, our cohort has not yet conducted any metabolomics-related research, and therefore, we are unable to provide relevant information. However, we are aware of the issue of limited sample size. Since 2023, we have established research cohorts in other regions of China, and in future studies, we will further collect information on dietary habits, infant growth and development, as well as conduct related omics research.

Reviewer 4 Report

Comments and Suggestions for Authors

The article has a scientific merit and provides readers with new data related to vitamins content in human milk and the potential change in its composition during storage. It connects to modern trends in science in order to use safe products, especially when we take into consideration infants and a good quality milk. There is not many research regarding that subject. In this case new approaches are needed and this article signs in that conception.

14: please explain the abbreviation HM in Abstract

Materials and methods

The methodology used is described in detail and is relevant to the research questions as well as is adequate and justified. In an experimental part the Author did apply modern analytical approaches, used high quality tools to do all research. These tools meet all criteria in current pre-clinical pharmaceutical sciences. However one question has arisen:

118: LC-MS/MS was done according to a standard procedure/protocol for these vitamins or modified by the Authors?

Results and Discussion:

- The results are presented in a clear and readable manner, statistics is included, the appropriate controls were used. The results have been discussed with those available in literature, relevant publications were cited. Generally, there are not many results in this subject and no many researchers deal with that.

Conclusion:

- please re-phrase this part, try to avoid a repetition of results etc. in this part. Make a conclusion, what more may be done in the future, how the milk can be used in the milk bank - what about this perspective?

Author Response

Comments 1: 14: please explain the abbreviation HM in Abstract

Response 1: Thank you for pointing this out. Agree. I have re-write this part. See L10

Comments 2: 118: LC-MS/MS was done according to a standard procedure/protocol for these vitamins or modified by the Authors?

Response 2: Currently, there is no standard method for determining the different forms of thiamine, riboflavin, and folate in HM. The method we adopted is based on previous research and analytical methods for HM and other biological samples, and was further developed by our team.

Comments 3: Conclusion:- please re-phrase this part, try to avoid a repetition of results etc. in this part. Make a conclusion, what more may be done in the future, how the milk can be used in the milk bank - what about this perspective?

Response 3: Agree. I have re-write this part. See L343-347

Round 2

Reviewer 1 Report

Comments and Suggestions for Authors

To the Authors

The Authors of the revised Ms ID nutrients-3447820 have, in my opinion, carefully addressed all my points of concern. Despite the already ascertained study limitations, the Ms is significantly improved in terms of presentation and impact to the scientific community.

Author Response

Thank you very much for your recognition. We will continue to conduct research in this area in the future.

Reviewer 2 Report

Comments and Suggestions for Authors

My previous comments were not addressed, especially comments 1, 2, 3, and 4. My 4th comment is particularly relevant.

Author Response

Comments 1: You should start your abstract by providing a background statement to justify the need for carrying out your study and then the study’s objectives need to be clarified. Finally, some directions for further investigation should be provided.

Response 1: Thank you for pointing this out. I have re-write this part again. See L9-10 and L30-31

Comments 2: References must be formatted according to the journal’s guidelines.

Response 2: Agree. I have formatted references. See L361-421

Comments 3: The Introduction is very brief and poor. Please, provide more information on the topics addressed in your study and you have to make clear the need to conduct the study. The inclusion of tables and illustrations in this section is recommended.

Response 3: Agree. I have re-write this part. See L65-69

Comments 4: Line 65: Why 35 pregnant women were recruited from only one center? How do you consider this sample representative and adequate?

Response 4: Firstly, due to the relative lack of systematic research on different forms of water-soluble vitamins in breast milk, it is difficult for us to accurately estimate the sample size. Of course, for China, which has 10 million newborns every year, recruiting only 35 pregnant women from one center lacks national representativeness. Therefore, as I explained for the first time, in order to ensure nationwide representativeness, we will continue to recruit more mothers from different regions in China to join the cohort. However, for the Pinggu District in Beijing, a relatively independent city with an annual birth population of less than 3,000, I believe a sample size of 35 is relatively sufficient for this area.